# Growth of *V. parahaemolyticus* in Tropical Blacklip Rock Oysters

**DOI:** 10.3390/pathogens12060834

**Published:** 2023-06-16

**Authors:** Anna C. Padovan, Alison R. Turnbull, Samantha J. Nowland, Matthew W. J. Osborne, Mirjam Kaestli, Justin R. Seymour, Karen S. Gibb

**Affiliations:** 1Research Institute for the Environment and Livelihoods, Charles Darwin University, Darwin, NT 0810, Australia; 2Institute of Marine and Antarctic Studies, University of Tasmania, Taroona, TAS 7053, Australia; 3Aquaculture Unit, Department of Industry, Tourism and Trade, Northern Territory Government, Darwin, NT 0801, Australia; 4Climate Change Cluster, University of Technology Sydney, Ultimo, NSW 2007, Australia

**Keywords:** Blacklip Rock Oyster, *Vibrio parahaemolyticus*, growth rate, food safety, temperature, tropical

## Abstract

The opportunistic pathogen *Vibrio parahaemolyticus* poses a significant food safety risk worldwide, and understanding its growth in commercially cultivated oysters, especially at temperatures likely to be encountered post-harvest, provides essential information to provide the safe supply of oysters. The Blacklip Rock Oyster (BRO) is an emerging commercial species in tropical northern Australia and as a warm water species, it is potentially exposed to *Vibrio* spp. In order to determine the growth characteristics of *Vibrio parahaemolyticus* in BRO post-harvest, four *V. parahaemolyticus* strains isolated from oysters were injected into BROs and the level of *V. parahaemolyticus* was measured at different time points in oysters stored at four temperatures. Estimated growth rates were −0.001, 0.003, 0.032, and 0.047 log_10_ CFU/h at 4 °C, 13 °C, 18 °C, and 25 °C, respectively. The highest maximum population density of 5.31 log_10_ CFU/g was achieved at 18 °C after 116 h. There was no growth of *V. parahaemolyticus* at 4 °C, slow growth at 13 °C, but notably, growth occurred at 18 °C and 25 °C. *Vibrio parahaemolyticus* growth at 18 °C and 25 °C was not significantly different from each other but were significantly higher than at 13 °C (polynomial GLM model, interaction terms between time and temperature groups *p* < 0.05). Results support the safe storage of BROs at both 4 °C and 13 °C. This *V. parahaemolyticus* growth data will inform regulators and assist the Australian oyster industry to develop guidelines for BRO storage and transport to maximise product quality and safety.

## 1. Introduction

Sydney rock oysters (SROs) (*Saccostrea glomerata*) and Pacific oysters (PO) (*Magallana gigas* (Thunberg 1793)) account for 99% of Australia’s oyster production (AUD 114M [1], with farms located in cooler temperate regions of New South Wales, South Australia, and Tasmania [2]. In contrast, Blacklip Rock Oysters (BROs) (*Saccostrea* lineage J) occur naturally in the Indo-Pacific region including across northern Australia [3] and are grown commercially on a limited scale. There is increasing interest in expanding the production of BROs in this region, particularly in remote Aboriginal communities, with current research focused on securing consistent spat supply and optimising production methods [4,5].

BROs grow in warm waters, which also support many species of indigenous aquatic microbes including *Vibrio* spp., which is potentially pathogenic to humans [6]. Among these, *V. parahaemolyticus* accounts for most seafood borne gastroenteritis [7] and is amongst the top emerging risks for food safety worldwide [8]. There is considerable global concern about the increasing incidences of seafood poisoning due to *Vibrio* blooms and warming sea temperatures in temperate regions [9,10,11,12,13]. In Australia, *Vibrio* spp. are recognised as an emerging food safety risk [14]. This threat to food safety and the emergence of a tropical oyster market for Australia means that there is an urgent need to learn as much as possible about *Vibrio*—BRO dynamics including the growth rates of potentially pathogenic *Vibrio* spp. at temperatures likely to be encountered post-harvest.

Post-harvest storage conditions are informed by recommendations to keep oysters as cool as possible to limit pathogen growth, while keeping the animals alive since dead seafood may lead to rapid spoilage and adversely affect microbiological safety [15]. The Australian Shellfish Quality Assurance Program [16] provides guidelines for postharvest practices to manage shellfish microbiological quality. The guidelines recommend shell stock intended for raw consumption to be cooled to 10 °C or less, within 24 h of harvest, unless there is evidence that higher temperatures will not support the unacceptable growth of human pathogens. POs and other shellfish are generally stored at these temperatures, but SROs are stored at 25 °C or less within 24 h of harvest and then at 21 °C or less within 72 h of harvest [17]. These guidelines are based on different responses of SROs and POs to spoilage at different temperatures, measured using aerobic plate counts and sulphide-producing bacteria [18] as well as different *V. parahaemolyticus* growth rates in these oyster species [18,19,20,21]. To calculate *V. parahaemolyticus* growth rates, researchers have used either oysters naturally infected with *V. parahaemolyticus* at the time of collection [22,23,24] or inoculated with a culture of *V. parahaemolyticus* [19,25], and measured *V. parahaemolyticus* inactivation or growth at different temperatures over time. The use of naturally infected oysters is more realistic but the large variability in the *V. parahaemolyticus* levels that may be present in individual oysters can make the interpretation of results difficult.

While recommended storage temperatures exist for POs and SROs, they may not be relevant for tropical oyster species that host a *Vibrio* community that is adapted to a tropical climate. The objective of this study was to determine the effect of storage temperature on the growth rate of tropical *V. parahaemolyticus* strains in artificially inoculated BROs and in doing so, provide the necessary foundation for postharvest temperature control plans for BROs.

## 2. Methods

### 2.1. Isolation of V. parahaemolyticus Strains from Oysters and Preparation of Inoculum

*V. parahaemolyticus* strains were isolated from BROs collected from the Tiwi Islands (S11.34097 E130.23645) and from milky oysters (*Saccostrea scyphophilla*) collected in Darwin Harbour (S12.33779, E130.908103), in the Northern Territory of northern Australia. During this isolation process, oysters were scrubbed under running potable water and shucked. The meat and liquor from 3 oysters were pooled, homogenised using an Ultra-Turrax^®^ IKA T18 (IKA^®^ Works, Rawang, Malaysia), diluted 1:1 (*w*/*v*) with 1× phosphate buffered saline (PBS; 10 mM phosphate buffer, 137 mM NaCl, 2.7 mM KCl, pH 7.4), and 100 µL was spread onto CHROMagar™ Vibrio (Dutec Diagnostics, St. Leonards, Australia) before the plates were incubated overnight at 35 °C. Mauve colonies typical of *V. parahaemolyticus* were picked and re-streaked onto fresh CHROMagar™ Vibrio twice more to obtain individual colonies.

Colonies were screened for *V. parahaemolyticus* by qPCR targeting the *tlh* [26] or *toxR* [27] genes using a pick and boil method to extract DNA from the plated colonies. Briefly, colonies were dispersed into 50 µL of sterile distilled water, boiled for 3 min, centrifuged at 13,500× *g*/10 min, and a 1 µL template was used in a qPCR assay. Isolates positive for the *tlh* or *toxR* gene were grown in tryptic soy broth (TSB) containing 2% (*w*/*v*) NaCl at 30 °C. Glycerol stocks of each isolate were prepared and stored at −80 °C. DNA from overnight cultures was extracted using the DNeasy^®^ UltraClean^®^ Microbial Kit (Qiagen, Hilden, Germany). A PCR assay targeting the *hsp60* gene [28] was performed on the extracted DNA, the amplicons purified using the ISOLATE II PCR and Gel Kit (Meridian Bioscience), and sequenced in both directions at the Australian Genome Research Facility. The forward and reverse sequences were assembled using MacVector v17.5.6 (MacVector Inc 2020) and the identities confirmed using BLAST (https://blast.ncbi.nlm.nih.gov/Blast.cgi, accessed on 15 June 2021). Four isolates were selected: M51 and M56 from BROs from the Tiwi Islands, and M116 and M117 from small milky oysters from Darwin Harbour. These isolates were also screened for the virulence genes *trh* and *tdh* [26], *vscC2*, *vopC*, and *vopP* [29], but all assays were negative (results not shown).

One day before the inoculation experiments started, bacterial isolates were streaked onto TSA with 2% NaCl and grown overnight at 30 °C. Four mL of sterile TSB/2% NaCl broth was inoculated with 2–3 individual colonies from each isolate separately and incubated with shaking at 30 °C for approximately 4–5 h until visibly turbid. The cultures were centrifuged at 10,000 rpm for 2 min and resuspended in filtered sterile seawater to give a final absorbance at 600 nm of between 0.15 and 0.25 units. Two mL from each of the 4 cultures were pooled to give the final inoculum. Serial dilutions were prepared using 1× PBS and 100 µL was plated onto TSA/2% NaCl to calculate the cell numbers, which were expressed as colony forming units (CFU) per mL.

### 2.2. Oyster Inoculation, Incubation and Processing

BROs (*Saccostrea* lineage J) were obtained from a commercial farm in Bowen (Queensland Australia) in two shipments of approximately 250 oysters each sent two weeks apart. Average seawater temperatures ranged from 23 °C to 25 °C at the time of sampling (https://data.aims.gov.au/aimsrtds/datatool.xhtml, accessed on 10 November 2022). The oyster shell length ranged from 49 to 74 mm with an average (± standard deviation) of 60 mm (± 6 mm). The first shipment of oysters was used for experiments at 4 °C and 13 °C and the second shipment was used for experiments at 18 °C and 25 °C. There was no significant difference in the background concentration of *V. parahaemolyticus* between shipments (3.56 ± 3.05 log_10_ CFU/g for the first shipment, 3.30 ± 2.87 log_10_ CFU/g for the second shipment; Welch’s *t*-test, *p* = 0.22). Oysters were placed in an open plastic bag in a polystyrene box and kept at 18 °C overnight. The following morning, the plastic bag was closed, ice bricks were added over a thick layer of newspaper, and the box sealed. The oysters were airfreighted to the Charles Darwin University laboratory in Darwin and the experiments commenced the next morning. On receipt, the temperature of the BROs was 15 °C and 18 °C in shipments 1 and 2, respectively.

Oysters were scrubbed and washed under running potable water. A 2–5 mm hole was made into the oyster lid approximately halfway along the length of the shell and 100 μL of either the filtered sterile seawater (control) or *V. parahaemolyticus* suspension was injected into the adductor muscle using a sterile 1 mL syringe fitted with a 22-gauge needle. The initial inoculum concentration was 2.0 × 10^7^ CFU/mL for the 4 °C and 13 °C experiment and 4.2 × 10^5^ CFU/mL for the 18 °C and 25 °C experiment. A higher concentration was used for the cooler temperatures to enable detection, as the levels were expected to decrease with storage.

Oysters were placed into open plastic bags in trays for storage in incubators set to 4 °C, 13 °C, 18 °C, and 25 °C. Temperature loggers were used to record the temperature. The four storage temperatures were chosen based on literature reviews, storage requirements in transporters and at seafood retailers, existing storage temperatures at harvest location, temperatures expected to be experienced during postharvest as well as consultation with the local BRO industry. Four degrees is within the temperature required (<5°C) for food safety practices, with transporter and retail chillers set to 1–5 °C. Thirteen degrees is close to the tipping point reported for *V. parahaemolyticus* growth and was an important temperature to determine the response of tropical compared to temperate *V. parahaemolyticus* isolates. Eighteen degrees is the temperature currently used by one commercial oyster grower for the short-term storage of oysters prior to transport, so it was a logical temperature to test. Finally, 25 °C is the approximate ambient dry season temperature for the Australian tropics and was chosen based on the perception that ambient temperatures, as opposed to refrigerated storage, are favourable for BROs with the industry aiming for a live product.

Five replicates were used for oysters injected with *V. parahaemolyticus*, with three oysters pooled per replicate. For the 4 °C and 13 °C experiments, the sampling times were 0, 24 h, 72 h, 120 h, 192 h, and 264 h. For the 18 °C and 25 °C experiments, the sampling times were 0, 12 h, 24 h, 72 h, 120 h and 168 h. The shorter experiment duration at the warmer temperatures was based on the expected reduced oyster viability compared to the cooler temperatures. The shell width of each oyster was measured, and the total meat and liquor weight of the pooled oysters recorded at time zero and at each time interval when the oysters were harvested. Twenty extra control and *V. parahaemolyticus* injected oysters were prepared and stored at each temperature to allow for losses during the experiment.

Controls were oysters injected with filtered sterile seawater, in duplicate with 5 oysters per replicate. Controls were sampled at the beginning, middle, and end of the experiment. The number of controls per replicate were to account for the expected variability in the background levels of indigenous *V. parahaemolyticus*. The controls primarily accounted for injuries sustained in the injection process and to track *V. parahaemolyticus* levels during the experiment. Gaping, non-responsive oysters were assumed to be dead and excluded from sampling.

At each time point, oysters were shucked, the meat and liquor pooled, and weight recorded. An equal volume of sterile 1× alkaline peptone water (APW, pH 8.4 (CM1028 Oxoid)) was added and the sample homogenised using an Ultra-Turrax^®^. The dispersion element was washed between replicates in the following sequence of solutions: potable water, 1% (*w*/*v*) Virkon™ disinfectant, potable water, 80% (*v*/*v*) ethanol, and sterile high pure water. On each sampling day, control oysters were processed before the *V. parahaemolyticus* injected oysters, and blanks were included (APW) to check for adequate tool disinfection. Serial dilutions of the homogenate were made in 1× PBS and 100 μL of each dilution plated in triplicate onto CHROMagar™ Vibrio. Plates were incubated at 30 °C overnight and mauve colonies (*V. parahaemolyticus*) counted and colony forming units (CFU) per gram oyster homogenate calculated. To confirm identity as *V. parahaemolyticus*, 50–60 mauve colonies were randomly picked and assayed by qPCR targeting the *tlh* gene as outlined above.

### 2.3. Analyses

Data were imported into Prism 9 for MacOS (GraphPad Software, LLC 1994–2021). Counts were transformed to log_10_ values, and lines or curves fitted to the data. Growth rates (log_10_ CFU/h) were calculated from the best fit lines at 4 °C and 13 °C. To calculate the specific growth rates (μ) and maximum population densities (log_10_ CFU/g) at 18 °C and 25 °C, data were imported into https://foodmicrowur.shinyapps.io/biogrowth/ (accessed on 14 October 2022) and fitted using a modified Gompertz model. Generalised additive models (GAMs) were fitted in R (version 3.6.0 2017-06-30; Copyright^© 2023^. The R Foundation for Statistical Computing) (library mgcv) to assess the nonlinear changes of *V. parahaemolyticus* counts over time. Models were fitted with a separate smooth term for time (hours) for each temperature group with temperature as an additional categorical predictor and using a negative binomial distribution (log link). A negative binomial generalised linear model (GLM) was also performed with the outcome *V. parahaemolyticus* counts and predictor temperature (categorical with four groups), a second-degree polynomial function for time (hours) as well as an interaction term between time and temperature. To evaluate whether *V. parahaemolyticus* growth significantly varied between different temperatures, the significance of interaction terms was assessed between temperature groups. Contrasts using the library emmeans were calculated based on the polynomial GLM to compare the estimated *V. parahaemolyticus* levels between temperature groups at 48 h and 72 h and using the Tukey method to adjust *p* values for multiple testing. To account for the nonlinear growth of *V. parahaemolyticus* at 18 °C and 25 °C, a second-degree polynomial function was fitted for time. All tests were 2-tailed and considered significant if *p* values were less than 0.05.

## 3. Results

### 3.1. V. parahaemolyticus Growth Rates in Injected Oysters

At the beginning of the experiment, after the initial inoculation of oysters with the *V. parahaemolyticus* cocktail, the concentrations and standard deviation of *V. parahaemolyticus* in the BROs were 5.001 ± 0.282 log_10_ CFU/g at 4 °C and 13 °C, and 3.567 ± 0.164 log_10_ CFU/g at 18 °C and 25 °C.

Changes in *V. parahaemolyticus* concentrations at 4 °C and 13 °C were best explained by a linear relationship (GAM model effective degrees of freedom (edf) 1.9, *p* < 0.001 for the latter), while at 18 °C and 25 °C, a curve best explained the data (Figure 1) (GAM model edf > 2; *p* < 0.001). No lag phase was observed in the growth curves. At 4 °C, there was no significant change in the *V. parahaemolyticus* levels over time (linear regression on log *V. parahaemolyticus* levels and GAM model *p* > 0.050), although the trend was a gradual decrease (Figure 1). At 13 °C, 18 °C, and 25 °C, there was a significant increase in *V. parahaemolyticus* levels over time (*p* < 0.001 for all models).

Estimated growth rates of *V. parahaemolyticus* in BROs were −0.001, 0.003, 0.032, and 0.047 log_10_ CFU/h at 4 °C, 13 °C, 18 °C, and 25 °C, respectively (Table 1). The highest maximum population density of 5.31 log_10_ CFU/g was achieved at 18 °C after 116 h.

Change in the *V. parahaemolyticus* levels over time varied significantly between all temperature groups (polynomial GLM model, interaction terms between time and temperature groups *p* < 0.05) except between 18 °C and 25 °C. At 48 h, there was a significant difference in *V. parahaemolyticus* concentrations only between 4 °C and 25 °C, but at 72 h, the *V. parahaemolyticus* concentrations significantly differed between all temperature groups with the exception of no difference between 18 °C and 25 °C (polynomial GLM model, *p* < 0.05) (Figure 2).

### 3.2. Control (Seawater Injected) Oysters

At the beginning of the experiment, concentrations (and standard deviation) of *V. parahaemolyticus* in the control BROs were 3.554 ± 0.136 log_10_ CFU/g at 4 °C and 13 °C from the first shipment of oysters, and 3.285 ± 0.167 log_10_ CFU/g at 18 °C and 25 °C from the second shipment (Figure 3).

Concentrations of *V. parahaemolyticus* initially decreased at 4 °C and 13 °C, but then increased again at the end of the storage period, after 11 days (257 h). Similarly, at 18 °C and 25 °C, the *V. parahaemolyticus* concentration decreased after 3 days (69 h), and then increased at day 5 (116 h). A final measurement was taken on day 7 (163 h) at 18 °C where the *V. parahaemolyticus* levels again decreased. The variability in concentrations was greatest at the warmer incubation temperatures.

## 4. Discussion

*Vibrio parahaemolyticus* seafood risk management is supported by implementing cold chain temperatures that minimise pathogen growth. Here, we present the first *Vibrio* risk data for BROs, which are the focus of a burgeoning aquaculture industry in northern Australia. Following injection into BROs, *V. parahaemolyticus* did not grow at 4 °C but grew at temperatures ≥13 °C. The tipping point for *V. parahaemolyticus* growth in oyster species is in the temperature range of 10–15 °C [19,22,25,30,31]. The low growth rate for *V. parahaemolyticus* in BROs reported in this study at 13 °C fits within this range and is notable given that the *Vibrio* strains used here were isolated from warm tropical waters. *V. parahaemolyticus* growth in BROs was minimal at 13 °C and significantly lower than the growth at warmer temperatures.

Our results show that storage of BROs at 4 °C will prevent the growth of *V. parahaemolyticus*, but since this storage temperature may also kill or impair these tropical oysters, the shelf life and quality at this temperature needs to be assessed in the event death accelerates spoilage by psychrotolerant microorganisms. At 13 °C, very low *V. parahaemolyticus* growth rates were measured in BROs, which may be a better temperature for BRO survival, however, this is not a standard commercial refrigeration temperature. Our study showed no significant difference in *V. parahaemolyticus* growth or maximum population densities in BROs at 18 °C or 25 °C, possibly because the oysters and their microbiome that adapted to these warmer temperatures are able to ‘manage’ introduced *V. parahaemolyticus* levels.

Compared with other oyster species, *V. parahaemolyticus* growth rates at 25 °C in Eastern oysters (*Crassostrea virginica*) and artificially inoculated Pacific oysters (PO) were higher than those measured in BROs (current study), which had similar rates to Asian oysters (*C. ariakensis*) (Figure 4). In addition, *V. parahaemolyticus* growth rates in POs, Eastern oysters, and Asian oysters increased with higher temperatures (25 °C compared to ~20 °C) (Figure 4), but this was not the case for BROs where there was no significant difference between growth at 18 °C and 25 °C. In contrast, *V. parahaemolyticus* did not grow in SROs stored at temperatures up to 28 °C [19,20,21], with growth only observed over 30 °C [20,32].

Variations in *V. parahaemolyticus* growth are often attributed to oyster immunology and their responses to substantial changes in their surrounds, the interaction of the introduced pathogen to resident oyster microbes, or the use of different experimental bacterial strains. Sydney rock oysters are considered a hardy species [15] and it has been suggested that lower microbial counts measured in SROs stored at 15 °C compared to 8 °C could be due to a more active immune system at the warmer temperature [18]. Intertidal molluscs have physiological and immunological adaptations to deal with conditions that can change quickly over a tidal cycle where they tolerate periods of emersion characterised by extremes in oxygen availability and temperature [33,34]. The type and extent of these responses [35,36] may influence their ability to cope with these stressors and subsequently impact their interaction with microbes [37,38]. Wild BROs are intertidal and are also considered a hardy species and may be better able to cope with substantial changes in their surrounds.

A recent study showed that virulent *V. parahaemolyticus* strains injected into *C. gigas* grew faster at 15 °C than the non-virulent strains [25]. In contrast, other studies using broth and *C. gigas* oyster slurry reported the more rapid growth of *V. parahaemolyticus* strains lacking the virulent *trh* gene compared to strains without *trh* [39]. Such comparisons between studies can be complicated by the use of different matrices as well as the use of different strains. In our study, a mix of four strains isolated from tropical rock oysters was injected in BROs to account for potential differences in growth between the strains. These four strains lacked both the *trh* and *tdh* genes, however, since vibriosis has been reported from *trh^-^/tdh^-^* strains [40], these markers are no guarantee of the capacity to cause disease. What constitutes a pathogenic strain is still the subject of much debate and whole genome sequencing is revealing new virulence factors [41,42] that contribute to infection. It is also possible that pathogenic strains respond differently in tropical BROs and the investigation of those strains in BROs will further our understanding of the behaviour of *V. parahaemolyticus* in stored tropical oysters.

Maximum *V. parahaemolyticus* population densities in oysters can vary by several orders of magnitude when stored at warmer temperatures. For example, maximum *V. parahaemolyticus* densities were higher in Eastern oysters stored at 20–25 °C [22] and PO injected with *V. parahaemolyticus*, but lower in natural POs and SROs [19] compared to BROs (current study). The maximum population densities for *V. parahaemolyticus* or any pathogen may depend on the type and density of other resident microbiota [43,44] including non-pathogenic environmental *Vibrio* species that may inhibit pathogenic *Vibrio* species [45,46]. Work is currently underway to measure the whole microbial community (total bacteria and *Vibrio* species) in stored BROs from this study to assess the impact of inoculated *V. parahaemolyticus* on the resident oyster microbiome compared to the seawater inoculated controls.

Due to the large natural variability in the *V. parahaemolyticus* levels in oysters, as evidenced by the seawater injected BRO controls and other reports [47], the approach used in this study was to inject a known number of cells into the oysters to avoid highly variable measurements between replicates and enable an accurate growth rate to be calculate. This also allowed for measurements of the *V. parahaemolyticus* levels at cooler temperatures following inactivation. Inoculation of oysters by filtration would better represent ingestion under natural conditions and be less invasive, however, because this can lead to variable uptake [25], the injection of bacteria was considered the most suitable inoculation method for this study.

Concentrations around 3.29–3.55 log_10_ CFU/g were measured before inoculating the BROs, which were at the higher end of the range reported in the temperate species POs and SROs [18,20,21,48]. *V. parahaemolyticus* is present almost year-round in the coastal seawater of northern Australia [6], with higher levels than more southern Australian locations [49]. There are periods of higher density in seawater, often related to season [6,50,51,52] or locations influenced by freshwater run-off [53], so it is therefore not unexpected that filter feeding organisms in the tropics may contain higher natural *V. parahaemolyticus* levels than their temperate counterparts. Storage at 18 °C and temperatures during transit may also have increased the natural levels of existing *V. parahaemolyticus* in the BROs in this study.

The oyster condition varies with season and is impacted by environmental factors such as algal blooms and oyster reproduction cycles in northern Australia [54]. These major physiological changes in oysters and their microbiome throughout their life cycle and seasons may alter their response to bacterial challenges, so the behaviour of *V. parahaemolyticus* in oyster tissue at various storage temperatures may vary depending on the oyster age and condition, which needs to be further explored.

## 5. Conclusions

In conclusion, these results support the storage of BROs at both 4 °C and 13 °C to minimise *V. parahaemolyticus* growth and set the foundation for regulators and the Australian oyster industry to develop storage and transport guidelines appropriate for tropical rock oysters to maximise product quality and food safety. Further post-harvest storage trials using pathogenic strains are required to determine if they respond differently to the non-pathogenic strains used in this study to further our understanding of the behaviour of *V. parahaemolyticus* in stored tropical oysters.

## Figures and Tables

**Figure 1 pathogens-12-00834-f001:**
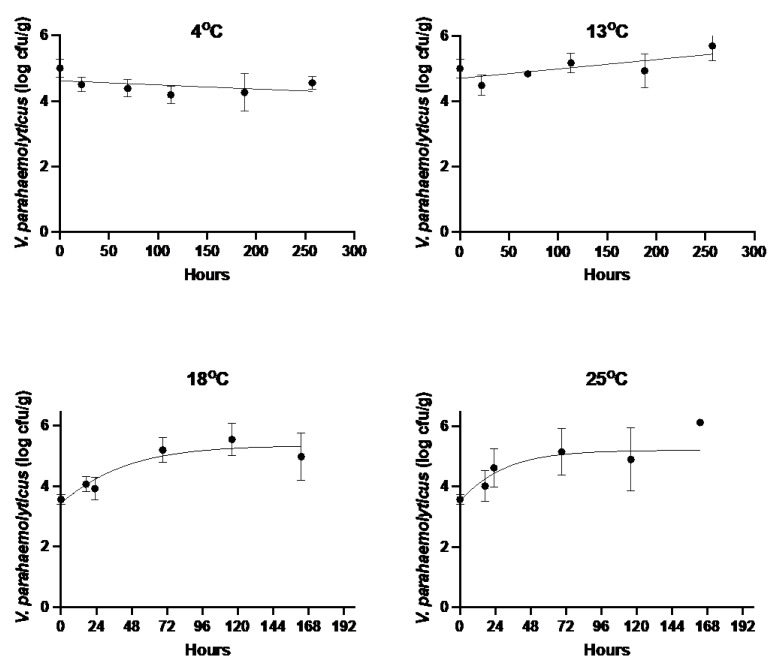
Growth profiles of *Vibrio parahaemolyticus* in Blacklip Rock Oysters stored at 4 °C, 13 °C, 18 °C, and 25 °C. Points indicate the averages of five replicates, bars are the standard deviation, and the lines indicate fitted curves. The last sample at 25 °C consisted of only one replicate.

**Figure 2 pathogens-12-00834-f002:**
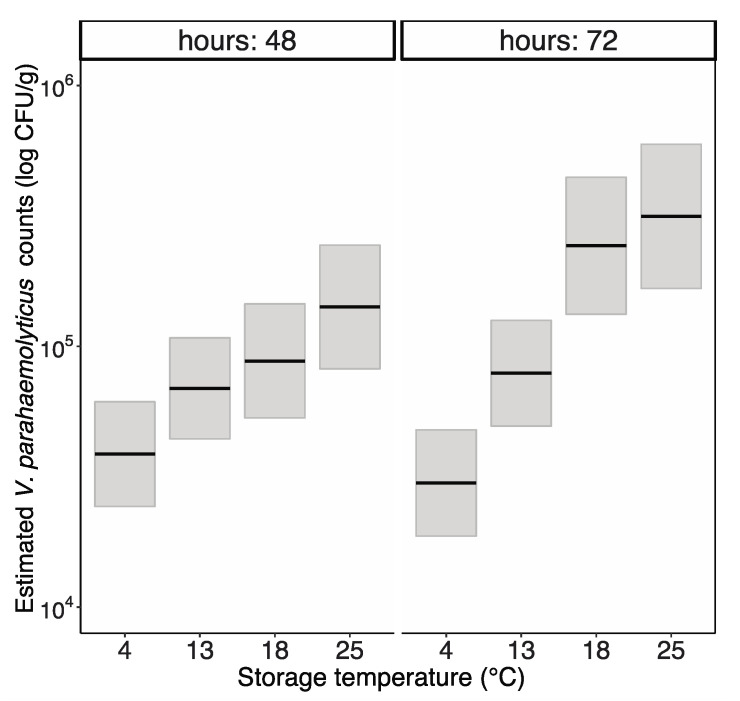
Estimated *Vibrio parahaemolyticus* growth in Blacklip Rock Oysters at each temperature after 48 and 72 h. Black line is the estimated average counts and the grey area is the 95% confidence interval based on a polynomial generalised linear model. Starting concentrations of *V. parahaemolyticus* injected into the oysters were 2.0 × 10^7^ CFU/mL for the 4 °C and 13 °C experiment and 4.2 × 10^5^ CFU/mL for the 18 °C and 25 °C experiment.

**Figure 3 pathogens-12-00834-f003:**
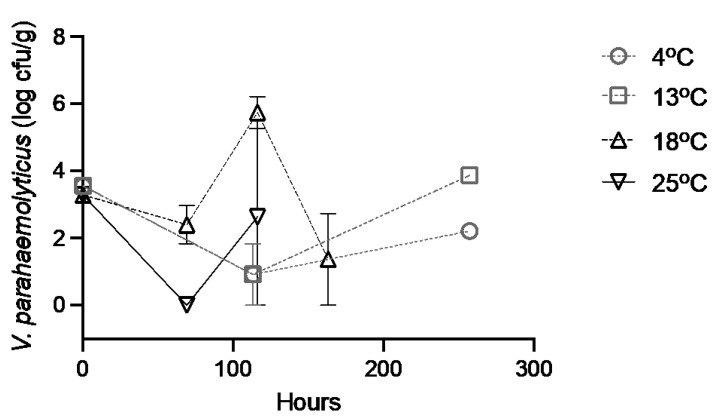
*Vibrio parahaemolyticus* concentrations in Blacklip Rock Oyster injected with filtered sterile seawater and stored at 4 °C, 13 °C, 18 °C, and 25 °C.

**Figure 4 pathogens-12-00834-f004:**
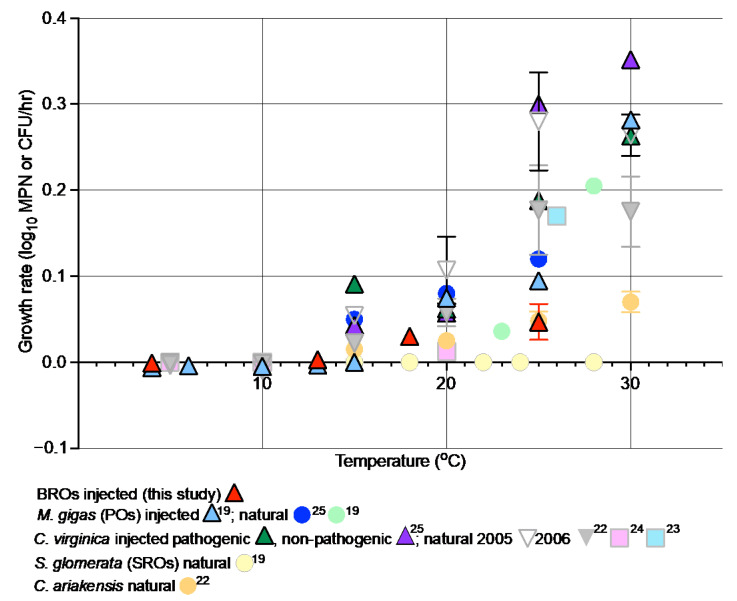
Growth rates of *Vibrio parahaemolyticus* in different oyster species at different temperatures [19,22,23,24,25].

**Table 1 pathogens-12-00834-t001:** Kinetic parameters for *Vibrio parahaemolyticus* growth measured over 264 h at 4 °C and 13 °C and over 168 h at 18 °C and 25 °C.

Storage Temperature (°C)	Growth Rate (Log CFU/h ± SE)	Maximum Population Density (log_10_ CFU/g ± SE)	Goodness of Fit (RMSE)
4	−0.0013 ± 0.0007	ND	0.390
13	0.0029 ± 0.0009	ND	0.408
18	0.032 ± 0.011	5.31 ± 0.245	0.463
25	0.047 ± 0.021	5.14 ± 0.394	0.652

Growth rates at 4 °C and 13 °C were calculated from the best fit lines. At 18 °C and 25 °C, the maximum specific growth rate (μ) and maximum population density were estimated from the modified Gompertz curves. ND, not determined. RMSE, root mean squared error.

## Data Availability

The data presented in this study are available on request from the corresponding author. The data are not publicly available due to lack of a suitable contextualised repository for bacterial growth data.

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
