# Peer review of "Growth of V. parahaemolyticus in Tropical Blacklip Rock Oysters"

_pathogens, 2023, doi:10.3390/pathogens12060834_

Round 1

Reviewer 1 Report

The manuscript titled “Growth of V. parahaemolyticus in tropical blacklip rock oysters (BRO)” described the growth characteristics of V. parahaemolyticus in BRO at various temperatures after post-harvest.      

The manuscript is easy enough to follow and is well written except for the discussion.  I have specific comments that need to be addressed before this manuscript is considered for publication.

In addition, please check the font and references throughout the manuscript again. Some studies in the manuscript are not found in the references  

please see the attached file 

Reviewer 2 Report

This article will provide valuable scientific evidence for helping properly preserve Blacklip Oysters during post-harvest, minor revisions could be done in some places:

1. What significance analysis was done to conclude that V. parahaemolyticus growth was not significantly different at 18 and 25 ℃, by among data of each time point among groups of samples? Or by comparing growth rate? Rephrase and clarify in the abstract, results and conclusion part.

2. In Line 142-145, there are two "potable water" and one "sterile high pure water", should double check.

3. Line 161, a dot "." missing after "Copyright (C) 2017".

4. Line 158 "growth rate (mu)" to "growth rate (μ)".

5. Line 188, "The last sample at 25 ℃ consisted of one sample only.", rephrase.

6. Line 206-207 "4.2 x 105 CFU/mL for the 4 ℃ and 13 ℃ experiment and 2.0 x 107 CFU/mL for the 18 ℃ and 25 ℃" to "2.0 x 107 for the 4 ℃ and 13 ℃ experiment and 4.2 x 105 CFU/mL for the 18 ℃ and 25 ℃"?

7. Line 108-109 "The first shipment of oysters were used for experiments at 4℃ and 13℃ and the second shipment was used for experiments at 18℃ and 25℃", include the initial V. parahaemolyticus in these oysters before inoculation, including if V. parahaemolyticus from these oysters of 2 shipments were significantly different.

8. As the strains tested in this study were non-pathogens, will this influence their variations in growth? Please include more explanation in the conclusion part. 

Reviewer 3 Report

The manuscript entitled “Growth of V. parahaemolyticus in tropical Blacklip Rock Oysters” is well written and has scientific merit. However, the authors are suggested to revise the manuscript for the following comments:

Introduction:

Before the end of the introduction, it is necessary to add information from the literature if there were any relevant previous studies.

Materials and Methods:

Line 100: please change 100µl à100µL

Lines 123-124: Why did the authors choose these four temperatures (4, 13, 18 and 25 ºC)? Based on the findings of this study, selecting 1-2 points between 4-13 ºC would be interesting.

Results:

-Line 192: Is it Table 1 or Table 2? The Table caption and Table position seem displaced. Please recheck.

Figure 3: The ºC for 18 and 25 ºC should not be superscript. 

Discussion:

In the caption of Figure 4, the reference numbers are missing; please check.

Conclusion:

As a storage temperature of 4 ºC may kill or impair tropical oysters, BROs storage at 4 ºC should not be recommended in conclusion. 

Overall Comments:

There are a few typos in the manuscript. Authors are recommended to revise the entire manuscript for typos. 

Reviewer 4 Report

This paper investigates the growth of Vibrio parahaemolyticus in the lesser used and studied Blacklip Rock Oysters.  The paper is very well written, clear, easy to understand and a pleasure to read.  The experimental design could be improved by investigating the four storage temperatures on the same batch of oysters and using a consistent inoculum.  Nonetheless, the study does provide useful information to the oyster industry and regulators.

Abstract

Well-written abstract.  I would suggest that Vibrio parahaemolyticus be shortened to V. parahaemolyticus when it is used mid-sentence after the first introduction of the species.  See line 17.

Line 22 What does “Results support BROs storage at both 4⁰C and 13⁰C” mean? Was the growth of the Vibrio low enough (at 13⁰C) that these temperatures were considered suitable for safe storage? The sentence seems truncated in order to meet the abstract word limit.  Perhaps reword to “Results support the safe storage of BROs at both 4⁰C and 13⁰C”, which should still be within the word limit.

Method

Well-written, comprehensive, easy to understand method.

Line 108 First shipment of oysters was used (shipment is singular, so the verb should be singular, ie was, not were).

I think it would have been preferable to have all four temperatures tested in each shipment.  This may not be the case, but it looks like the first two temperatures were conducted with one shipment, and then two more temperatures were conducted with a second shipment after the results of the first experiments were known.  I understand the rational behind the two inoculum levels.  For consistency, I think it would have been better to conduct the trials using one inoculum level, especially as growth could still occur at 13⁰C and that is harder to measure with a high starting inoculum.  It also weakens the conclusions regarding comparisons between the temperatures.

Results

At 13⁰C, the growth could have been greater if the initial inoculum had not been as high.  There wasn’t much room for growth when starting at 107.

There are at times large error bars in the growth profiles, particularly at later timepoints.

This is understandable considering that the oysters will be variable and are living.  It does weaken the conclusions, particularly at 25⁰C.

Line 228 “Vibrio strains used here were isolated from warm 228 tropical waters.”  Although this is a valid comment, presumably mentioned to indicate why growth was less at 13⁰C, the sentence is in a different font and seems out of place at this point in the paragraph.  Please incorporate it into the paragraph more seamlessly.  

Discussion

The discussion is well-written, interesting and informative.

Conclusion

The conclusion is valid.

Line 322 – should be “Formal analysis”
